# Crystal Structure of the Catalytic Domain of a Botulinum Neurotoxin Homologue from *Enterococcus faecium*: Potential Insights into Substrate Recognition

**DOI:** 10.3390/ijms241612721

**Published:** 2023-08-12

**Authors:** Kyle S. Gregory, Peter-Rory Hall, Jude Prince Onuh, Otsile O. Mojanaga, Sai Man Liu, K. Ravi Acharya

**Affiliations:** 1Department of Life Sciences, University of Bath, Claverton Down, Bath BA2 7AY, UK; kg540@bath.ac.uk (K.S.G.); peterrory.hall@bioch.ox.ac.uk (P.-R.H.); jp.onuh@bath.edu (J.P.O.); oom21@bath.ac.uk (O.O.M.); 2Protein Sciences Department, Ipsen Bioinnovation Limited, 102 Park Drive, Milton Park, Abingdon OX14 4RY, UK; sai.man.liu@ipsen.com

**Keywords:** botulinum neurotoxin, *Enterococcus faecium*, homologue, light chain, autoproteolysis, AlphaFold

## Abstract

*Clostridium botulinum* neurotoxins (BoNTs) are the most potent toxins known, causing the deadly disease botulism. They function through Zn^2+^-dependent endopeptidase cleavage of SNARE (soluble N-ethylmaleimide-sensitive factor attachment protein receptor) proteins, preventing vesicular fusion and subsequent neurotransmitter release from motor neurons. Several serotypes of BoNTs produced by *Clostridium botulinum* (BoNT/A-/G and/X) have been well-characterised over the years. However, a BoNT-like gene (homologue of BoNT) was recently identified in the non-clostridial species, *Enterococcus faecium,* which is the leading cause of hospital-acquired multi-drug resistant infections. Here, we report the crystal structure of the catalytic domain of a BoNT homologue from *Enterococcus faecium* (LC/En) at 2.0 Å resolution. Detailed structural analysis in comparison with the full-length BoNT/En AlphaFold2-predicted structure, LC/A (from BoNT/A), and LC/F (from BoNT/F) revealed putative subsites and exosites (including loops 1–5) involved in recognition of LC/En substrates. LC/En also appears to possess a conserved autoproteolytic cleavage site whose function is yet to be established.

## 1. Introduction

Botulinum neurotoxins (BoNTs) are produced by the anaerobic spore-forming bacteria *Clostridium botulinum* and cause neuromuscular paralysis (i.e., botulism) by proteolytic cleavage of SNARE (soluble N-ethylmaleimide-sensitive factor attachment protein receptor) proteins [1]. Eight serologically distinct BoNTs from *Clostridium botulinum* (BoNT/A-BoNT/G, and BoNT/X) have been identified [2,3,4] that vary in receptor [5] and substrate recognition [6]. All BoNTs are expressed as a single polypeptide chain that must be cleaved post-translationally to form the active di-chain molecule which is comprised of a 100 kDa heavy chain (HC) and 50 kDa light chain (LC). They all share a similar three-dimensional structure; the heavy chain folds into two distinct domains (the cell binding domain [H_C_] and the translocation domain [H_N_]), and the LC is a Zn^2+^-dependent metalloprotease (Figure 1) [7,8,9]. Extending perpendicular from the central H_N_ domain is a loop known as the ‘belt’ that wraps around the LC—this has been reported to act as both chaperone and pseudo-substrate in BoNT/A [10]. The BoNT mechanism of toxicity involves three steps, occurring with respect to each of these domains [11]. Initially, the H_C_ recognises both the protein and ganglioside receptor on the surface of motor neurons [12] (except for BoNT/C which recognises two ganglioside receptors [13]). BoNT/A [14], BoNT/D [15], BoNT/E [16] and BoNT/F [17] bind isoforms of synaptic vesicle glycoprotein 2 (SV2A/B/C), and BoNT/B [18] and BoNT/G [19] recognise synaptotagmin (I/II). 

Upon binding, BoNT is internalised via receptor-mediated endocytosis [23], and acidification of the endosome is believed to cause a conformational change in the H_N_ domain [24,25,26] to allow passage of the LC into the cytosol. Once in the cytosol, the LC is free to cleave its target SNARE protein, inhibiting acetylcholine release from motor neurons. BoNT/A and BoNT/E cleave the 25 kDa synaptosomal protein (SNAP-25) [27]; BoNT/B [28], BoNT/D [29], BoNT/F [30], BoNT/G [31], and BoNT/X [32] cleave the vesicle-associated membrane protein (VAMP); and BoNT/C cleaves both SNAP-25 [33] and syntaxin [34]. This highly specific and potent mechanism has led to the exploitation of BoNTs as therapeutics for the treatment of both hyper muscular and endocrine disorders. Currently only BoNT/A and BoNT/B are used as therapeutics [35].

BoNT-like genes have also been identified in non-clostridial species such as *Weissella oryzae* (BoNT/Wo) and *Enterococcus Faecium* (BoNT/En). The BoNT/Wo gene was isolated from fermented rice grain used for livestock feed [36,37], whereas BoNT/En was isolated from cow faeces [38,39]. The presence of BoNT-like genes in these species could, therefore, pose a potential health threat to both humans and livestock, although their functional activity and host target(s) remain to be established. A BoNT-like gene in *E. faecium* may be a particular concern because of the species’ effective ability at horizontal gene transfer. Indeed, *E. faecium* is the leading cause of hospital-acquired multi-drug resistant infections through the transfer of antibiotic-resistant genes [40]. The low abundance of *E. faecium* strains within the BoNT/En gene cluster may be due to its recent acquisition [39]; therefore, it is important to understand its molecular features. This information will guide our understanding of BoNT evolution and aid in the development of potential effective therapies in the event that BoNT-like genes are disseminated amongst gut microbiota.

The BoNT/En gene shares 29–38.7% sequence identity with other BoNTs. It is most similar to BoNT/X and contains the key conserved features of the BoNT family, a SxWY ganglioside binding motif within the H_C_ domain, a cysteine pair which links the H_N_ and LC, and a conserved HExxH zinc-coordinating motif [38]. Interestingly, BoNT/En possesses the unique ability to cleave at least one member of all three SNARE-protein families (SNAP-25, VAMP, and STX); however, the cleavage of STX1 was only observed in vitro [38]. The cleavage of SNAP-25 and SNAP-23 occurs at the N-terminal end, between Lys69 and Asp70 (for SNAP-25), which is distinct from all known BoNT SNAP-25 cleavage sites. The N-terminal fragment of SNAP-25 was undetected through mass spectrometry (LC-MS/MS) following cleavage, suggesting possible further degradation by LC/En [39]. The VAMP cleavage sites for all BoNT serotypes are located within a 25 amino acid residue stretch, with LC/En cleaving VAMP1, VAMP2, and VAMP3 between an Ala and Asp residue (Ala69 and Asp70 for VAMP1, Ala67 and Asp68 for VAMP2, and Ala54 and Asp55 for VAMP3). This scissile bond is just one residue away from the cleavage site of its most closely related homologue (BoNT/X), which cleaves between an Arg and Ala residue (Arg68 and Ala69 for VAMP1, Arg66 and Ala67 for VAMP2, and Arg53 and Ala54 for VAMP3). Similarly, the reported cleavage sites for BoNT/F and BoNT/D (which share 33.3% sequence identity) are only a single residue apart.

An understanding of how BoNT/En is able to recognise a variety of substrates would allow for the future development of inhibitors and antibodies towards BoNT/En and also its utilisation as a potential therapeutic. Here, we present the crystal structure of LC/En at 2.0 Å resolution and perform a detailed structural analysis in comparison to the previously reported LC/A and LC/F structures in complex with SNAP-25a and a VAMP-like inhibitor, respectively, providing possible insights into the substrate recognition mechanism of LC/En.

## 2. Results and Discussion

### 2.1. Crystal Structure of LC/En

Initial attempts at LC/En crystallisation in a protein buffer containing 50 mM Tris pH 7.4, 200 mM NaCl, and 1 mM DTT produced poorly diffracting crystals across several crystallisation screens. A thermal shift assay [41,42] was used to identify a more stabilising protein buffer (0.144 M HEPES pH 7.5, 1 mM TECEP), which yielded high-quality crystals. The best crystals grew in 0.1 M SPG buffer (succinic acid, sodium phosphate monobasic monohydrate, and glycine) pH 7.0, 25% *w*/*v* PEG 1500 and diffracted to high resolution. The structure of LC/En was determined at 2.0 Å with one molecule in the asymmetric unit (Table 1) by molecular replacement using the predicted structure from the AlphaFold2 server [20] as a search model. Clear electron density was observed throughout the structure except for residues 127–130 and 249–260 which could not be modelled. The overall fold of LC/En (Figure 2A) is similar to all BoNT serotypes and homologues identified to date [6,32,43], consisting of a mixture of α-helices (31%) and β-strands (19%).

### 2.2. The Active Site and Subsites

The catalytic site presents the tetrahedral geometry typical of the BoNT family of metalloprotease domains [45]. The Zn^2+^ ion is coordinated by the HExxH motif residues, His225, Glu226 (via a bridged phosphate ion that has likely replaced the catalytic water molecule as observed in BoNT/A structure [46]), His229, and an additional residue, Glu269 (Figure 2A, inset). These residues are conserved across all BoNT serotypes and their homologues, with differences in substrate specificity occurring due to variation in several exosites within an extended cleft [47]. This cleft is known to span the circumference of the molecule (Figure 3A,C,D) and is occupied by the ‘belt’ in full-length BoNT structures [7,9]. This is similarly true for the AlphaFold2-predicted structure of BoNT/En (Figure 1). To assess the implication of differences at the active site clefts of BoNT serotypes, the cleft volumes for LC/En, LC/A, LC/F, LC/Wo, and LC/X were calculated (using the program’s 3 V cavity, channel, and cleft volume calculator [44]). The analysis indicates that LC/En appears to have a much larger cleft volume than its closest related homologue, LC/X, and is most similar to LC/A (Figure 2B). As BoNT/En is known to cleave SNARE proteins from all three SNARE-protein families, it is probable that it possesses either reduced specificity within its exosites and subsites or additional exosites compared to other BoNTs.

The putative LC/En subsites (Table 2 and Figure 3A,B) were determined by comparison of LC/En with the LC/A structure in complex with an uncleavable SNAP-25 mimic [48]. Variations within these subsites (compared to LC/A) may provide insight into why LC/En appears to be a much more promiscuous SNARE-protein protease. The presence of a glycine residue within the putative S1 subsite is particularly interesting as it would contribute to enhanced flexibility and reduced steric hindrance close to the catalytic zinc ion, such that it is less restricted by which substrate it can accommodate. Additionally, the lysine residues at positions 168 and 195 in place of hydrophobic residues (Phe163 and Phe194 in LC/A) within the S1′ subsite may further contribute to substrate promiscuity, given their increased length, rotamer angles, and potential ability to form both polar and non-polar contacts within the active site depending on the substrate. Although LC/En is known to cleave SNAP-25 at a site distinct from that of LC/A, the properties of residues in the S3′ and S5′ subsites appear to be mostly conserved (Table 2). The equivalent residue of Tyr367 (which forms part of S3′) and Arg364 (which forms part of S1′ and S2′) (Tyr366 and Arg363 in LC/A) are known to stabilise the reaction transition state during substrate catalysis in LC/A [49] and are therefore likely to perform a similar role in LC/En.

### 2.3. Putative Substrate Binding Exosites

We performed a detailed structural comparison of LC/En to the previously determined LC/A and LC/F structures in complex with SNAP-25a (LC/A:SNAP-25) [47] (residues 141–204) and a VAMP2-based inhibitor (LC/F:VAMP2) [50], respectively. Additionally, LC/En was superimposed with the full-length BoNT/En AlphaFold2-predicted structure to identify regions of the cleft involved in binding of the belt [20]. These analyses allow for the identification of putative LC/En exosites for the recognition of both SNAP-25 and VAMP2, as the belt is known to mimic substrate binding [10]. 

The belt interacts with LC/En (from the BoNT/En AlphaFold2-predicted structure) via 37 residues forming a total of 22 hydrogen bonds and 2 salt bridges. Superimposition of the lone belt (from the BoNT/En AlphaFold2-predicted structure) with the LC/En crystal structure, reveals 17 potential hydrogen bonds and 2 salt bridges. These results were combined to generate a list of all possible interactions between LC/En and the heavy-chain belt (Table 3). Residues that are clustered in 3D space (Figure 3A,C,D) and that interact with the heavy-chain belt offer insight into the location of possible exosites. 

Additionally, we have identified a total of five loops (designated as loops 1–5) (Figure 4) that have been associated with substrate and belt recognition in the LC/En, LC/A:SNAP-25, and LC/F:VAMP2 structures. Relative to the scissile bond, LC/A loops 1, 2, 3 and 5 bind the N-terminal portion of SNAP-25, whereas loop 4 forms both the S3′ and S5′ subsite to recognise the C-terminal portion [47]. These loops vary in both length (Table 4) and conformation (Figure 4A) across the serotypes accounting for variation in substrate binding. Given the proximity of the VAMP cleavage sites for LC/En and LC/X, their substrate recognition sites are likely to be spatially similar. The scissile bond of VAMP2 for LC/F is nine residues away from the scissile bond for LC/En, indicating that the subsites will vary, and previously loops 1, 2 and 3 were identified as exosites for VAMP binding in LC/F [50].

Loops 1 and 4 are significantly longer in LC/En compared to LC/A, LC/F, and LC/X (Table 4). The additional portion of loop 1 contains several charged residues whereas loop 4 is predominantly hydrophilic (Figure 4B), and the increased length of both loops may contribute to the substrate promiscuity observed for LC/En. Superimposition of LC/En with the LC/A:SNAP-25 and LC/F:VAMP2 structures reveal that SNAP-25 and VAMP2 would sterically clash with loop 1 in LC/En (Figure 4C) and that the heavy-chain belt is positioned ~6 Å away from both SNAP-25 and the VAMP2. However, in the full-length BoNT/En AlphaFold2-predicted structure, loop 1 adopts a wider conformation that extends to form an interaction with the heavy-chain belt (Figure 4C). This suggests a difference in substrate positioning involving loop 1 compared to LC/A and LC/F and provides some insight into potential dynamics unique to LC/En due to its increased length. In LC/A, loop 4 is five residues shorter (Table 4) and along with loop 360–371 is referred to as the β-exosite (Figure 5). This exosite contributes to the formation of the S3′ and S5′ subsites (Table 2, Figure 3A,B), which are essential for binding SNAP-25 at its C-terminal end [48,51]. Loop 4 has been shown to alter conformation upon substrate binding, and its importance was recently highlighted in a study that probed the protease activity using single-domain antibodies. The antibodies target the S1′-S5′ subsites (including loop 4) and adopt a similar conformation to that of SNAP-25 at this site [52]. It is, therefore, probable that loop 4 serves a similar function in LC/En, stabilising the P’ end of SNAP-25 and VAMP, due to its location relative to the active site zinc ion (Figure 5A). Interestingly, LC/Wo also has a comparably long loop 4 structure containing one additional residue (compared to LC/En) (Figure 5B) and has been suggested to possess wider substrate specificity [43]. 

Despite shorter sequence lengths in loops 2, 3, and 5 of LC/En, the interactions with the heavy-chain belt region indicate that they may still be involved in substrate recognition. Part of loop 2 was identified as a putative exosite, as residues 131–135 form a β-sheet stacking interaction with residues 499–504 of the heavy-chain belt (Figure 4D), which is further anchored by hydrogen bonding (Table 3). Loop 2 is longer in LC/A and LC/F (Table 4) and forms hydrophobic contacts with the substrates (Val 129^LC/A^, Val 131^LC/F^). The equivalent residue in LC/En (Met 133^LC/En^) forms an intramolecular hydrogen bond with Arg 153^LC/En^ and is, therefore, unlikely to contribute to substrate recognition. In the LC/A and LC/F structures, loop 2 forms a β-sheet stacking interaction with loop 5 instead, which forms an exosite for the recognition of both SNAP-25 (LC/A) and VAMP2 (LC/F). The LC/En heavy-chain belt is recognised by loop 5, and may, therefore, contribute to substrate recognition; however, loop 2 and loop 5 are structurally distinct with loop 5 positioned some 14 Å away relative to LC/A loop 5 (Figure 4D). The shortness of loops 2 and 5 in LC/En, compared to LC/A and LC/F, prevents the formation of secondary structures between them, which may increase the flexibility of this region. Loop 3 interacts with the heavy-chain belt through backbone β-sheet interactions with residues 507–509, which also binds loop 1 through hydrophobic contacts (Table 3). Both loop 1 and loop 3 flank the active site (Figure 4A), forming what has been referred to as exosite 1 and 2 in LC/F. As the heavy-chain belt of BoNT/A mimics the substrate well within this region [47]), LC/En most likely will bind SNAP-25 and VAMP similarly to how it appears to bind the heavy-chain belt in the AlphaFold2-predicted full-length structure. Taken together, loops 1 and 3 may account for the difference in substrate binding close to the active site in LC/En. 

To explore the dynamics of these loop regions, we performed a normalised B-factor analysis using the web tool ‘BANΔIT’ [53] of LC/En, LC/A, LC/F, and LC/X (Figure 6) due to different resolutions for the available crystal structures. Despite the significant increase in the length of loop 1, LC/En appears slightly more dynamic than LC/A and LC/F; however, it is important to note that crystal contacts could contribute to the stabilisation of these loops within individual crystal structures. These plots in general support our structural analysis, revealing an increase in flexibility within loop 2 of LC/En due to the loss of its secondary structure, as well as a more dynamic loop 4 compared to LC/A LC/F and LC/X. The appearance of larger B-factors in loop 4 of LC/X is likely due to the absence of residues 249–260 in the LC/En crystal structure. Furthermore, the low B-factor variation in loop 5 indicates its increased length (in LC/A and LC/F compared to LC/En) does not significantly contribute to the enhanced flexibility of this exosite. We attribute this to the secondary structure observed between loops 2 and 5 in LC/A and LC/F (Figure 4D), which is absent in LC/En.

### 2.4. The α-Exosite

In addition to the β-exosite and five loop regions identified above, an α-exosite is also reported in the LC/A:SNAP-25 structure. The α-exosite consists of four helices, that upon SNAP-25 binding, form a 5-helix bundle with a helical segment of SNAP-25 (Figure 7A). The formation of the 5-helix bundle is driven by hydrophobic forces, and there is clear conservation of hydrophobicity within the four helices across the serotypes (Figure 7B). The length of the minimum SNAP-25 fragments required for cleavage by LC/A, LC/E, and LC/C suggests that the α-exosite is essential for the stabilisation of the N-terminal end of its substrates [27]. Given that SNAP-25 is entirely helical in the SNARE complex structure, upon binding LC/En, it will likely adopt a helical structure within the putative α-exosite, which is structurally conserved. Additionally, the heavy-chain belt forms a partial helical structure close to the putative α-exosite further supporting this hypothesis. Although the length of the inhibitor does not permit the identification of whether VAMP2 adopts a helical structure within the α-exosite, the terminus is positioned close to the α-exosite. As VAMP is also predominantly helical in structure in the SNARE complex, it is possible that it will adopt a similar 5-helix bundle with LC/En for stabilisation. 

### 2.5. LC/En Undergoes Autoproteolysis

To assess LC/En stability, samples were stored at room temperature for 1 h, 3 days, and 4 days (prior to and after subjection to heat stress and freeze—thaw) and analysed by SDS-PAGE (Figure 8A). With an increase in time, two bands corresponding to protein fragments at ~30 kDa and ~20 kDa (Figure 8A) appear, indicative of either protein degradation or autoproteolysis. 

Several autoproteolysis sites have been observed previously in LC/A, LC/B, and LC/E [54,55]. Interestingly, only one cleavage site appears to be structurally conserved, consistently occurring within loop 4. The corresponding cleavage within loop 4 of LC/En would result in the production of protein fragments at ~30 kDa and ~20 kDa (Figure 8B). Furthermore, immunoblotting using an anti-polyhistidine antibody produces a signal for the ~50 kDa intact LC/En (containing an N-terminal His tag) and the ~30 kDa N-terminal fragment, but not the ~20 kDa C-terminal fragment of LC/En (Figure 8A). There are an additional two bands present on the western blot, which is indicative of further degradation/cleavage products, but given their absence on the SDS-PAGE, their concentration suggests cleavage at these sites is not as significant. These results support the premise that LC/En undergoes autoproteolysis within loop 4. In the LC/En structure, loop 4 could not be fully modelled, and whether it is cleaved could not be structurally determined. However, as the majority of the protein remains at ~50 kDa on the polyacrylamide gel, the LC/En structure is of the uncleaved protein. Previous attempts at separating the C and N-terminal fragments of LC/A in the folded protein proved unsuccessful [56,57], suggesting that the cleaved LC maintains its 3-dimensional structure. In an attempt to determine the precise cleavage site within loop 4, we performed intact electrospray ionisation mass spectrometry (ESI-MS). The results of ESI-MS indicate the presence of several fragments which do not correspond to the molecular weight expected for the intact protein, which we suspect occurs due to either fragmentation during electrospray ionisation or additional degradation/cleavage sites. To address the effect of random fragmentation of the intact protein and that of the ~30 kDa N-terminal fragment and ~20 kDa C-terminal fragment, we performed trypsin digestion of LC/En. Trypsin selectively cleaves proteins at the carboxy end of lysine and arginine residues, resulting in several peptide fragments which start and end with a lysine or arginine residue that can be detected by ESI-MS. The results of trypsin-digested LC/En ESI-MS provided 72.8% sequence coverage (Figure 9) and identified three fragments with sequence coverage within loop 4 (205–262, 211–262, and 253–283). Fragments 205–262 and 211–262 arise due to cleavage by trypsin as they both follow and end in a lysine residue, whereas fragment 253–283 follows an asparagine and ends with a serine residue. Therefore, fragment 253–283 is not a result of trypsin digestion. Along with the SDS-PAGE, Western blot, and presence of a ‘conserved’ autoproteolytic site within the equivalent ‘loop 4′ of LC/A, LC/B, and LC/E, this supports the premise that LC/En may also contain an autoproteolytic site within loop 4.

The function of C-terminal autoproteolysis in LC/A has been identified as a requirement for complete activation of the LC, as the C-terminus was shown to interact near the active site [58]. The function within loop 4, if any, is unknown, but it has been suggested that it may increase the LC’s flexibility to aid in translocation or substrate recognition [55]. Given the inactivity of LC prior to release from the heavy-chain belt [10], the latter is more convincing, but it is interesting to speculate on the possibility that released LC may aid in the translocation of other BoNT molecules through autoproteolysis. The location of the autoproteolytic site is most intriguing due to loop 4’s importance in substrate turnover [48]. Whether autoproteolysis occurs in a natural system has not been well-studied, and it may simply be due to the high concentrations used to study LC in vitro. However, the presence of a conserved autoproteolytic site in a distant BoNT homologue suggests an evolutionary advantage which requires further investigation. 

## 3. Materials and Methods

### 3.1. Cloning, Expression, and Purification of LC/En

Codon-optimised full-length BoNT/En DNA was provided by Ipsen Bionnovation Ltd. LC/En was subcloned into the pOPINF vector via KpnI and HindIII restriction sites in frame with the hexahistidine tag using the inFUSION cloning kit (Takara Bio, Tokyo, Japan). The gene was expressed in Lemo21 (DE3) competent *E. coli* cells (NEB, Ipswich, MA, USA)which were grown in TB media supplemented with Ampicillin at 37 °C, 225 rpm, until an OD_600_ of 0.6. The temperature was reduced to 16 °C and protein expression was induced with 1 mM IPTG for 18 h. Cells were harvested by centrifugation at 5000 RCF, 4 °C, for 30 min. Cell pellets were resuspended in 5 mL/g of 50 mM Tris pH 7.4, 0.5 M NaCl, 10 mM imidazole supplemented with cOmplete EDTA-free protease inhibitors and lysed by homogenisation using the constants cell disrupter system at 20 kpsi twice. Lysate was clarified by centrifugation at 20,000 RCF, 4 °C for 30 min. Supernatant was collected and protein was captured using a 5 mL HisTrap HP column at 5 mL/min. Protein was eluted via a gradient with 50 mM Tris pH 7.2, 0.5 M NaCl, and 0.5 M imidazole. Protein was desalted into 50 mM Tris pH 7.4, 15 mM NaCl, and purified by QHP anion exchange using a 5 mL HiTrap Q HP column and eluted via a gradient with 50 mM Tris pH 7.2, 1 M NaCl. Protein was further purified by hydrophobic interaction chromatography; firstly, the concentration of AmSO_4_ of the sample was raised to 1 M, and protein was captured using a HiTrap Phenyl HP column. The protein was then eluted via a gradient with 50 mM Tris pH 7.4. LC/En was then further cleaned, and buffer exchanged into 0.144 M HEPES pH 7.5, 1 mM TECEP by gel filtration using a GE Healthcare superdex 200 column. Purified protein was flash-frozen in liquid nitrogen and stored at −20 °C until required for crystallisation. LC/En was concentrated to 6 mg/mL for preliminary crystallisation trials and verified by SDS-PAGE and Western blot using a monoclonal anti-polyhistidine peroxidase antibody. All reagents were sourced from Merck (Gillingham, UK), unless otherwise stated.

### 3.2. Crystallisation and Structure Determination

LC/En was concentrated to 12 mg/mL using a 10 kDa centrifugal concentrator. Protein crystallisation conditions were identified using the sitting-drop vapour diffusion method with high-throughput screens (Molecular Dimensions, Rotherham, UK) using the Art Robbins Phoenix crystallisation nano dispenser. The best crystals of LC/En were produced in a 1:1 reservoir: protein ratio at 16 °C in 0.1 M SPG (succinic acid, sodium phosphate monobasic monohydrate, and glycine) buffer, pH 7.0, 25% *w*/*v* PEG 1500. A total of 3600 images were collected at 100 K, with 0.1° of oscillation, 0.0044 s of exposure time, and an estimated dose of 10.00 MGy. Data were indexed and integrated in DIALS [59], scaled and merged in AIMLESS as part of the CCP4 suite [60], and initial phases were estimated by molecular replacement using phaser [61] and AlphaFold2 server [20]. The structure was refined using REFMAC5 [62] and Coot [63] and validated using Molprobity [64] and PDB validation. Figures were produced using ccp4mg [65]. Sequence alignments were produced using Clustal Omega [66] and ESPript [67].

### 3.3. Trypsin-Digested Electrospray Ionisation Mass Spectrometry (ESI-MS) of LC/En

A total of 100 µL of LC/En (at ~3 mg/mL) was used for trypsin digestion. An amount of 200 µL of 10 mM DTT in 50 mM ambic pH 8.4, was added to the sample and incubated at 56 °C for 30 min. The sample was left to cool at room temperature before addition of 200 µL of 55 mM iodacetic acid in 50 mM ambic pH 8.4 and incubated in the dark for 30 min. Then, 20 µL of trypsin (25 ng of trypsin in 50 mM ambic pH 8.5) was added at room temperature for 15 min, followed by further incubation at 37 °C overnight. The following day, 50 µL of 0.1% trifluoroacetic acid was added, followed by a further 50 µL after 10 min. The sample was then dried using the SpeedVac vacuum to a final volume of ~20 µL prior to ESI-MS analysis. The Agilent QTOF 6545 with Jetstream ESI spray source coupled to an Agilent 1260 Infinity II Quat pump HPLC with 1260 autosampler, column oven compartment, and variable wavelength detector (VWD) was used for ESI-MS of trypsin-digested LC/En. Positive ionisation mode was used with a gas temperature of 325 °C, the drying gas at 13 L/min, and the nebulizer gas at 35 psi (2.41 bar) in the 100–2000 m/z ranges collecting 5 spectra/sec. The sheath gas temperature was set to 300 °C and flow to 12 L/min. For MS/MS, the mass ranges were 50–2000 m/z collecting 3 spectra/sec with an isolation width set to medium (4 amu). Ions with charge states of 2, 3, or more were selected for fragmentation. For 2+, the slope was 3.1 with an offset of 1, 3+ the slope was 3.6 with an offset of −4.8, and for more than 3 the slope was 3.6 with an offset of −4.8. Ten precursors were selected per cycle, actively excluded after 3 spectra for 0.2 min. The MS was calibrated using a reference calibrant introduced from the independent ESI reference sprayer. Chromatographic separation was performed on a Water Acquity BEH C18 2.1 × 50 mm, 1.7 µm using H20 (Merck, LC-MS grade) with 0.1% formic acid (FA, Fluka) *v*/*v* and acetonitrile (ACN, VWR, HiPerSolv) with 0.1% FA *v*/*v* as mobile phase A and B, respectively. The column had a total run time of 12 min, where the flow rate was set at 0.3 mL/min at 50 °C, starting with 1% mobile phase B for 0.5 min; afterwards, the gradient was set to 5 min at 40% B, then 100% B at 7 min, and then held at 100% B for 2 min before being returned to 1% B at 9.1 min. The VWD was set to collect 280 and 320 nm wavelengths at 2.5 Hz. Finally, 10 µL injections of the samples were made [68]. Data processing was automated in BioConfirm v 10 (Build 10.01.10136).

## 4. Conclusions

A detailed structural comparison of the crystal structure of LC/En to the full-length BoNT/En AlphaFold2-predicted structure, and the previously reported LC/A and LC/F structures in complex with SNAP-25 (residues 141–204) and a VAMP2-based inhibitor, revealed putative LC/En substrate binding subsites and exosites, including loops 1–5. Features such as an increase in sequence length at loops 1 and 4 offer an insight into substrate binding. Loop 1 is expected to recognise the P end of its substrates, given its proximity to the active site zinc ion. The putative S3′ and S5′ subsites are partly formed by loop 4, which was also shown to be the location of an autoproteolytic cleavage site conserved across LC/A, LC/B, and LC/E. Loop 4 is, therefore, predicted to have an important role in the stabilisation of the P’ end of its substrates, and autoproteolysis may aid in substrate binding. Additionally, the increased flexibility of loops 1, 2, and 4 (compared to LC/A and LC/F) could account for the substrate promiscuity observed in LC/En. Furthermore, the heavy-chain belt (from the AlphaFold2-predicted structure) occupies a position in LC/En that is distinct from that of both the heavy-chain belt and substrate of LC/A and LC/F, due to variation with loops 1, 2, 3, and 5. As LC/En recognises at least one SNARE protein for all three SNARE-protein families, an initial inhibitor design could make use of SNAP, VAMP, and syntaxin mimics. Finally, the identification of these putative subsites and exosites provides important molecular information which may aid in the development of potent LC/En inhibitors; however, further experimental work is required to fully understand the substrate recognition mechanism of LC/En.

## Figures and Tables

**Figure 1 ijms-24-12721-f001:**
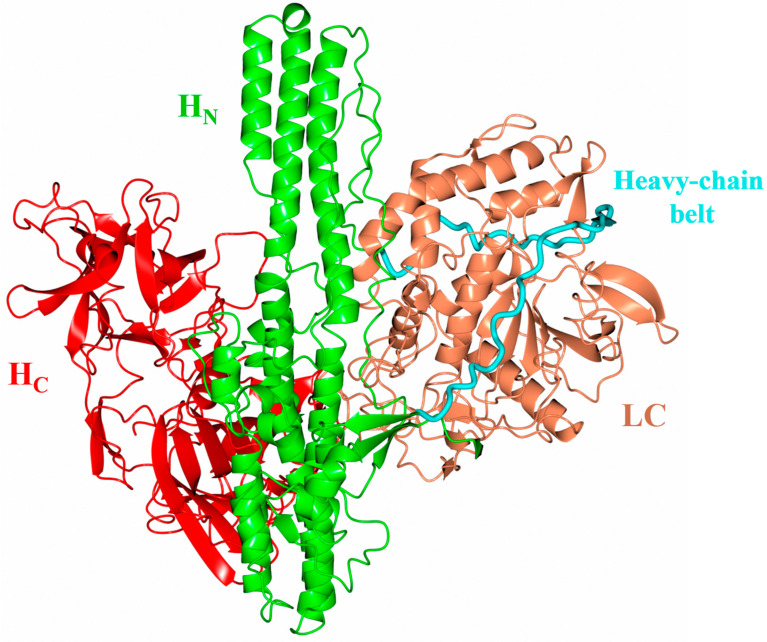
Botulinum neurotoxin structure. Alphafold2 [20] predicted structure of full-length BoNT homologue from *Enterococcus faecium* indicating the overall three-dimensional structure. This structure appears to be conserved across BoNT and its homologues. The cell binding domain (H_C_, red) and translocation domain (H_N_, green) make up the heavy chain, which is connected to the light chain (LC, orange) via a single disulphide bond. Extending perpendicular from the central H_N_ domain is the heavy-chain belt (cyan) which wraps around the LC acting as a pseudo-substrate. As the structure illustrated here is an AlphaFold2 prediction, the positioning of the domains relative to one another may be inaccurate. The domain organisation of the BoNT/En AlphaFold2-predicted structure appears to be ‘in between’ the two domain organisations determined experimentally. In the crystal structure of BoNT/A [7] and BoNT/B [21], the domain organisation is described as an ‘open-wing’ conformation [5,22] with all three domains in the same plane. In the BoNT/E structure, the H_C_ domain and LC domain display a ‘closed-wing’ conformation [8].

**Figure 2 ijms-24-12721-f002:**
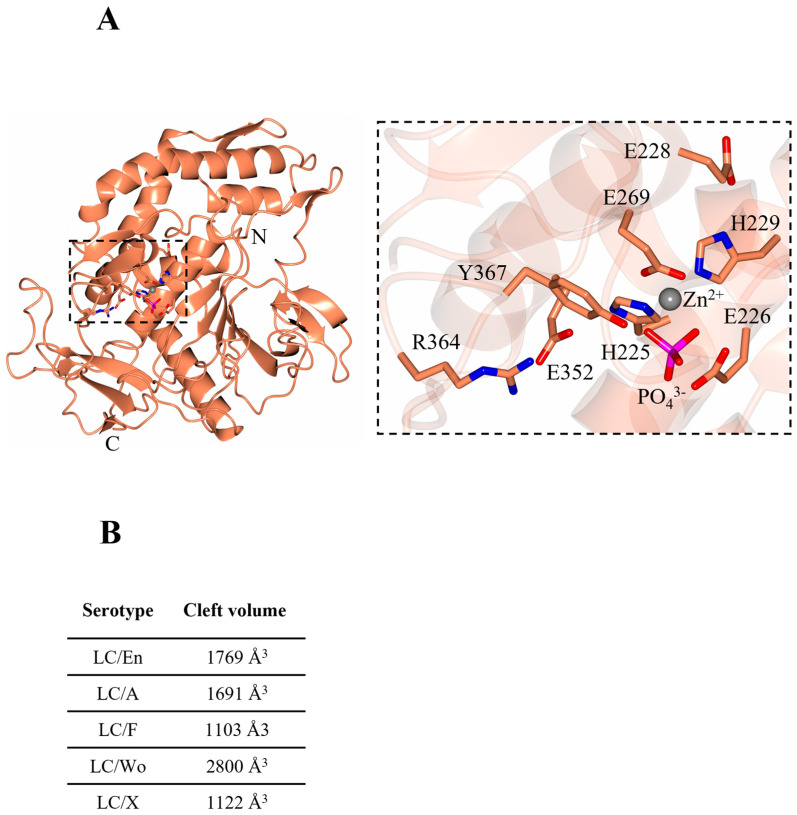
BoNT-like (BoNT homologue) light chain from *Enterococcus faecium*. (**A**) Crystal structure of BoNT-like light chain (LC) from *Enterococcus faecium*. The inset highlights the catalytic residues of LC/En, which are conserved across BoNT serotypes. (**B**) cleft volume sizes across BoNT serotypes/En/A,/F,/Wo, and/X were calculated with a 3 Å probe size using the 3 V cavity, channel, and cleft volume calculator [44]. The following PDB coordinates were used: LC/En (present structure), LC/A (2IMC), LC/F (2A97), LC/Wo (6RIM), and LC/X (6F47).

**Figure 3 ijms-24-12721-f003:**
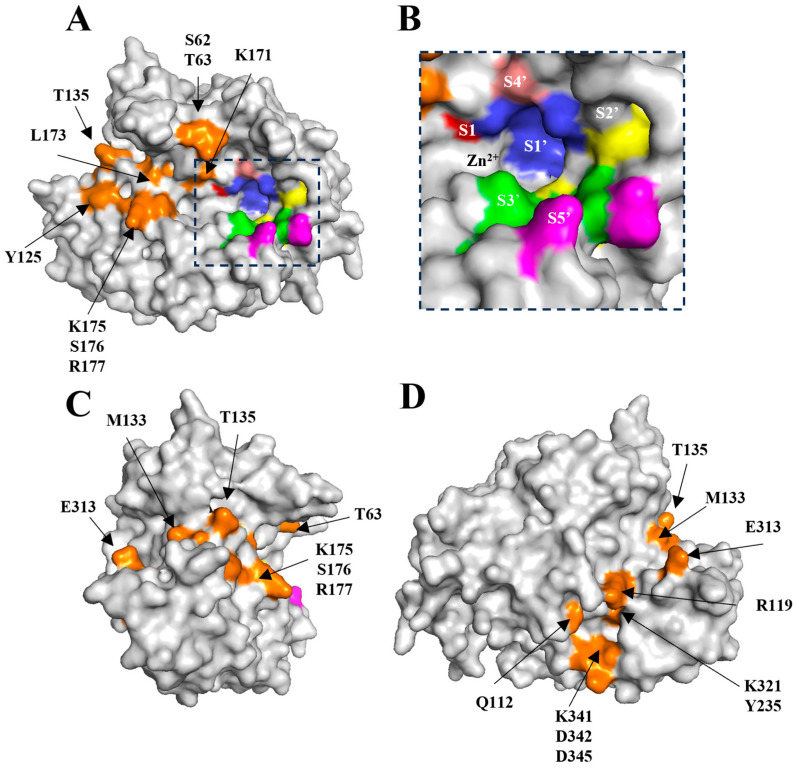
Location of putative subsites and exosites of LC/En. (**A**) Surface representation of LC/En with subsites and exosites highlighted. (**B**) Putative LC/En subsites. (**C**) Vertical 90° rotation of (**A**). (**D**) Vertical 180° rotation of (**A**). The putative subsites are coloured red (S1), blue, (S1′) yellow (S2′), green (S3′), pink (S4′) and magenta (S5′). The putative exosites are coloured orange and extend throughout the entire cleft.

**Figure 4 ijms-24-12721-f004:**
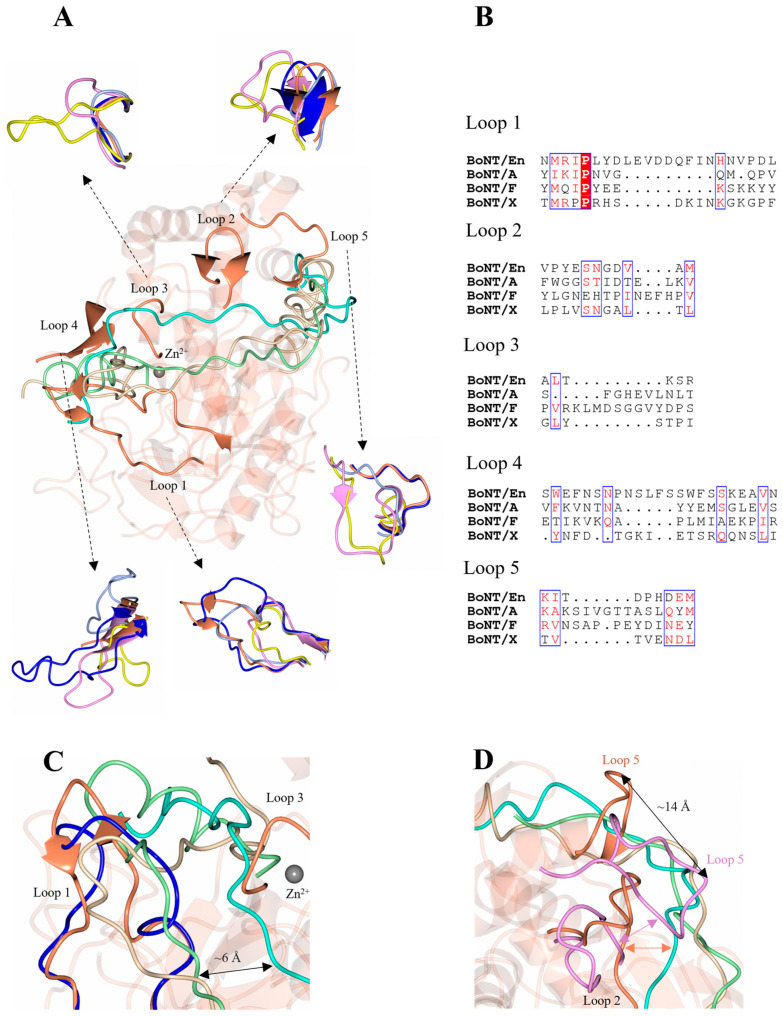
Putative LC/En substrate binding exosites. (**A**) Five variable loop regions were identified as SNAP-25, VAMP2, and heavy-chain belt recognition sites among LC/En, LC/A, LC/F, and LC/X. (**B**) Sequence alignment of the variable loop regions across LC/En, LC/A, LC/F, and LC/X. Loop 1 and loop 4 are longer compared to LC/A, LC/F, and LC/X. Conserved residues are shown in red text with a white background. The blue box indicates regions of similarity. Non-conserved residues are shown in black text with a white background. (**C**) Loop 1 and loop 3 positions relative to SNAP-25 (from LC/A structure in complex with SNAP-25), the VAMP2-based inhibitor (from LC/F in complex with a VAMP2-based inhibitor), and BoNT/En heavy-chain belt (full-length AlphaFold2-predicted structure). (**D**) Comparison of loop 2 and 5 in the LC/A and LC/En (LC/F loops are omitted from the figure for clarity) structures. In LC/A and LC/F, loops 2 and 5 interact through a β-sheet interaction (illustrated by pink arrow) forming an exosite for SNAP-25 and VAMP2 recognition. In LC/En, loops 2 and 5 are positioned away from each other, with loop 5 approximately 14 Å away relative to loop 5 of LC/A, and LC/En appears to form a β-sheet interaction with the belt instead (illustrated by orange arrow). LC/En is shown in orange, LC/A in magenta, LC/F in yellow, LC/X in grey, and BoNT/En (AlphaFold2-predicted structure) in blue. The SNAP-25 substrate is shown in beige, the BoNT/En heavy-chain belt (AlphaFold2-predicted structure) in cyan, and the VAMP2-based inhibitor in green.

**Figure 5 ijms-24-12721-f005:**
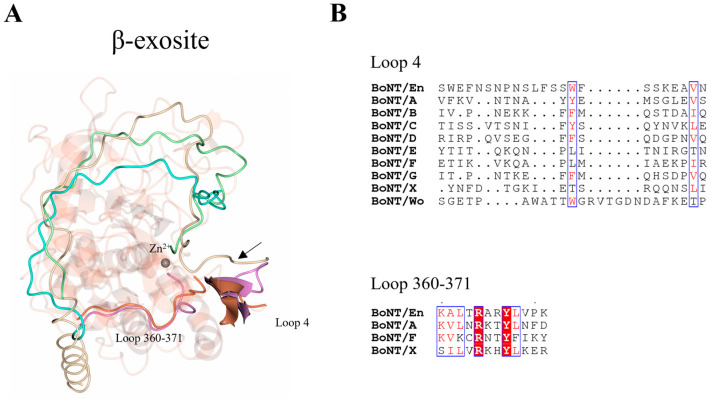
Location of β-exosite. (**A**) The β-exosite was previously identified using the crystal structure of LC/A (magenta), consisting of loop 4 and the 360–371 loop (LC/En residue numbering). The structurally equivalent regions for LC/En (orange) are shown, highlighting potential substrate recognition sites. Positional differences of SNAP-25, BoNT/En heavy-chain belt (from AlphaFold2-predicted model), and LC/F are shown by superimposition with LC/En. In LC/A, SNAP-25 forms a β-sheet stacking interaction with loop 4 (illustrated by the black arrow) and additional interactions with the 360–371 loop. The β-sheet stacking interaction with loop 4 is likely conserved across all serotypes, as this is the only structural region capable of stabilising the P’ end of the substrate. (**B**) Sequence alignment of loop 4 (from all serotypes) and the 360–371 loop (for LC/En, LC/A, LC/F, and LC/X). Conserved residues are shown in red text with a white background. The blue box indicates regions of similarity. Non-conserved residues are shown in black text with a white background. SNAP-25 is shown in beige, BoNT/En H_N_-belt (AlphaFold2 structure prediction) in cyan, and the VAMP2-based inhibitor in green.

**Figure 6 ijms-24-12721-f006:**
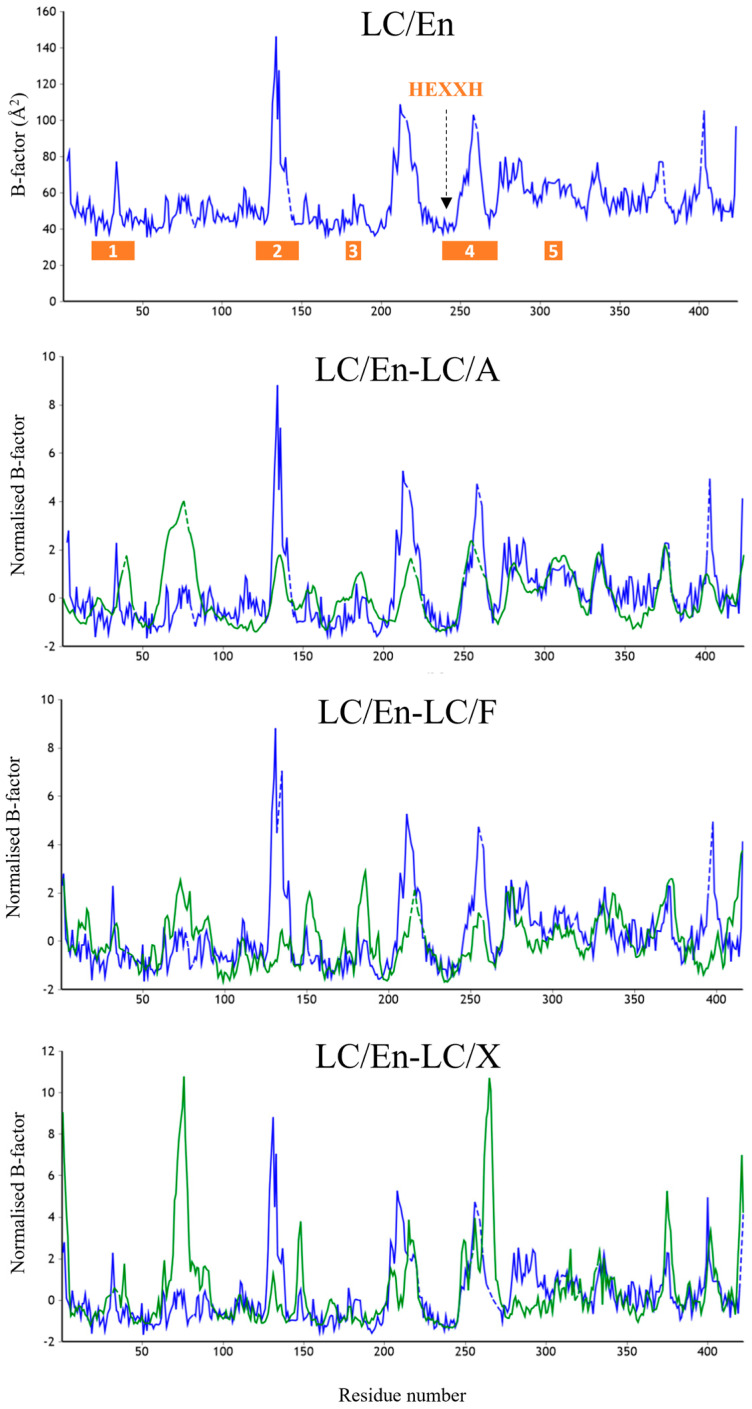
B-factor analysis of LC/En, LC/A, LC/F, and LC/X. Raw B-factor analysis of LC/En, and normalised B-factor comparison of LC/En-LC/A, LC/En-LC/F, and LC/En-LC/X. The analysis was performed using the web tool ‘BANΔIT’ [53]. For each normalised B-factor comparison plot, LC/En is shown in blue, and LC/A, LC/F, and LC/X are shown in green. Loops 1–5 are shown by the orange bar in the raw B-factor plot of LC/En.

**Figure 7 ijms-24-12721-f007:**
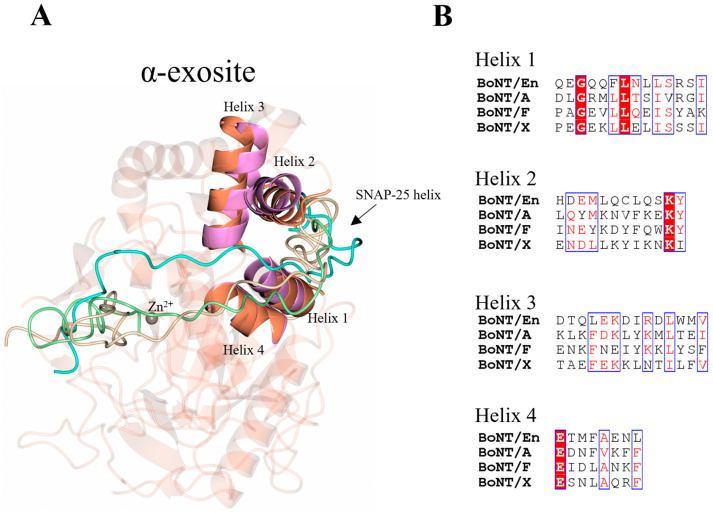
Location of α-exosite. (**A**) The α-exosite was previously identified using the crystal structure LC/A (magenta). The structurally equivalent regions for LC/En are shown (orange), highlighting potential substrate recognition sites for LC/En. In LC/A, a portion of SNAP-25 forms a 5-helix bundle with the α-exosite, which is likely to form upon LC/En substrate recognition also as both SNAP-25 and VAMP are predominantly helical in structure. (**B**) Sequence alignment of the α-exosite helices across LC/En, LC/A, LC/F, and LC/X. Conserved residues are shown in red text with a white background. The blue box indicates regions of similarity. Non-conserved residues are shown in black text with a white background. SNAP-25 is shown in beige, the BoNT/En heavy-chain belt (AlphaFold2-predicted structure) in cyan, and the VAMP2-based inhibitor in green.

**Figure 8 ijms-24-12721-f008:**
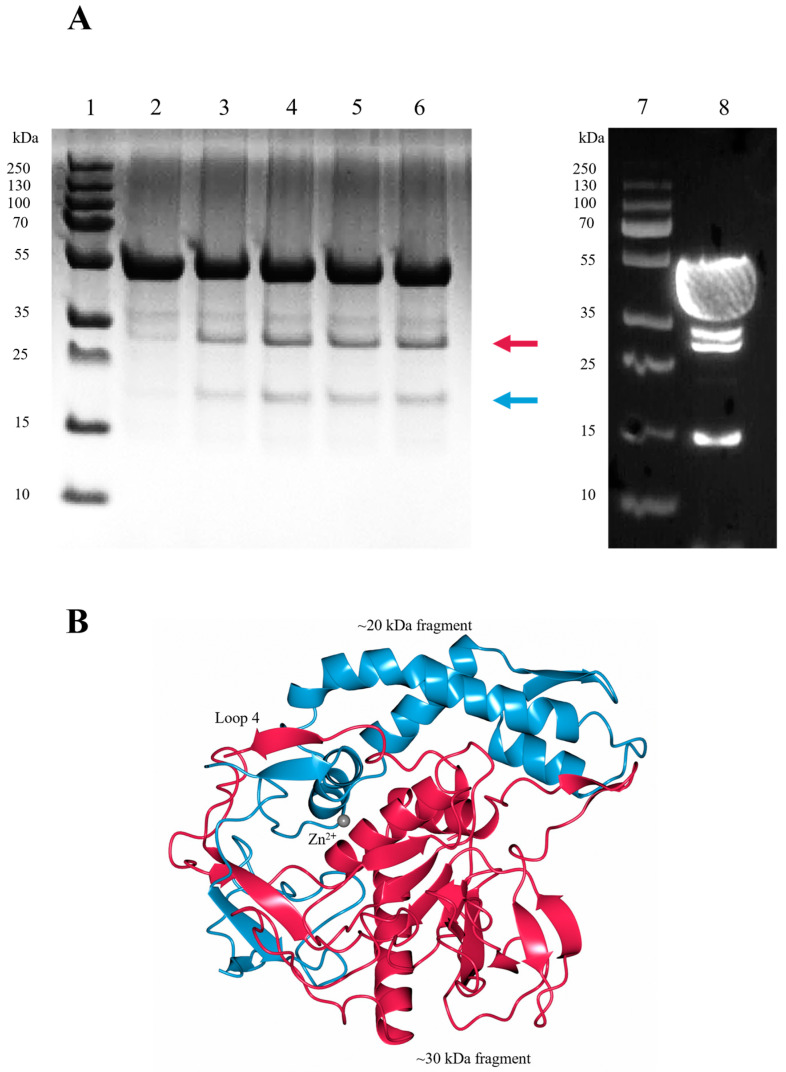
Autoproteolytic cleavage of LC/En. (**A**) SDS-PAGE (lanes 1–6) and Western blot (lanes 7–8) analysis of LC/En. SDS-PAGE was performed after incubation at room temperature for 1 hour (lane 2), 3 days (lane 3), 4 days with subjection to heat stress, (lane 4) 4 days (lane 5), and 4 days with subjection to freeze–thaw (lane 6) following size exclusion. Western blot was performed on LC/En that was concentrated to 6 mg/mL for preliminary crystallisation trials (lane 8). PageRuler Plus protein ladder is shown in lanes 1 and 7. The red arrow indicates the ~30 kDa fragment, and the blue arrow indicates the ~20 kDa fragment, and only N-terminal fragments are visible by Western blot analysis, indicating the presence of the 30 kDa fragment, along with the intact 50 kDa LC/En and further LC/En fragments at ~34 kDa and ~15 kDa indicative of other degradation/cleavage sites. (**B**) LC/En colour coded to represent the ~30 kDa (red) and ~20 kDa (blue) fragments that would arise from autoproteolytic cleavage at loop 4, which is consistent with the autoproteolytic cleavage site observed in LC/A, LC/B, and LC/E.

**Figure 9 ijms-24-12721-f009:**
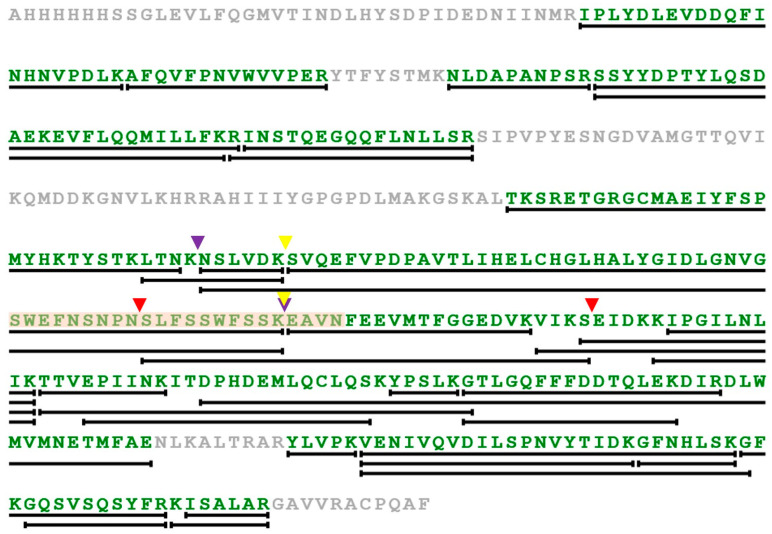
Electrospray ionisation mass spectrometry (ESI-MS) sequence coverage map of trypsin-digested LC/En. To identify fragments that arise from both trypsin and non-trypsin digestion we performed ESI-MS. Residues identified during ESI-MS peptide screening that equate to a total sequence coverage of 72.8% are shown in green. The lines underneath regions of the sequence indicate fragments identified during ESI-MS. Loop 4 is highlighted in light orange. Fragments that occur within loop 4 are illustrated by coloured arrows, with fragment 205–262 shown by the purple arrows, fragment 211–262 by the yellow arrows, and fragment 253–283 by the red arrows.

**Table 1 ijms-24-12721-t001:** X-ray crystallographic data collection and refinement statistics for LC/En. Outer shell statistics are in parentheses.

Beamline	I04 (Diamond Light Source)
Wavelength	0.9537 Å
Calculated dose	10.0 MGy
**Crystallographic statistics**	
Resolution (Å)	101.75–2.0 (2.05–2.0)
Space group	P 4_2_2_1_2
Unit cell dimensions	
a, b, c (Å)	143.894, 143.894, 51.695
α, β, γ (°)	90.00, 90.00, 90.00
No. reflections	982,082
No. unique reflections	37,319 (2683)
Completeness (%)	100 (100)
*R_merge_*	0.131 (3.877)
*R_pim_*	0.036 (0.763)
<I/σ(I)>	13.2 (0.9)
CC_1/2_	1.00 (0.626)
Multiplicity	26.3 (26.6)
**Refinement statistics**	
R_work_/R_free_	0.19/0.23
RMSD bond length (Å)	0.0106
RMSD bond angles (°)	1.910
Ramachandran plot statistics (%)	
Favoured	97%
Allowed	3.00%
Outliers	0.00%
Average B-factors (Å2)	
Amino acids	58.78
Zinc	48.17
Ligands	69.23
Water	54.73
Loop 1	50.04
Loop 2	90.04
Loop 3	52.84
Loop 4	80.84
Loop 5	72.09
Number of atoms	
Amino acids	3226
Zinc	1
Ligands	24
Water	156

**Table 2 ijms-24-12721-t002:** Putative LC/En subsites. The LC/A subsites were previously identified by co-crystallisation of LC/A with a SNAP-25-based inhibitor [48]. The equivalent residues for LC/En have been aligned, indicating potential SNAP-25 and VAMP2 subsites. Subsites S3′ and S5′ are essential for substrate recognition in LC/A. Given the conservation in residue properties at subsites S3′ and S5′, these subsites are likely involved in the stabilisation of all substrates’ P’ ends. Residues which are conserved or defined as conservative/semi-conservative substitutions based on polarity are shown in green to illustrate the conservation of residues within the subsites.

	S1	S1′	S2′	S3′	S4′	S5′
**LC/En**	G169	R364 T222 M166 V372 P217 K168 K195	R364 V369 V372	W258 E263 V265 Y367 V369 P370	A167	W258 E263 S260 L442 P370
**LC/A**	E164	R363 T220 I161 D370 T215 F163 F194	R363 N368 D370	Y251 L256 V258 Y366 N368 F369	Q162	Y251 L256 S254 F423 F369

**Table 3 ijms-24-12721-t003:** LC/En and BoNT/En (AlphaFold2 structure prediction) interactions with the heavy-chain belt. List of possible hydrogen-bonding interactions present across the LC/En (present structure) and BoNT/En heavy-chain belt (AlphaFold2-predicted structure) interface, as determined by superimposition of LC/En with the full-length AlphaFold2-predicted structure. Salt bridges are shown in italics. K171 forms a salt bridge with D513 in the present structure, and D511 in the AlphaFold2-predicted structure. Residues clustered together in the 3-dimensional structure may provide insight into potential LC/En exosites.

LC/En (Residues 1–425)	Belt (BoNT/En residues 484–514)
Q112	L484
K341	S485
D342	N486
D345	
K321	Y487
Y235	
R119	D488
R119	K491
E313	T496
E313	S497
M133	P500
T135	I502
M133	
T135	V504
S176	D506
Y125	
K175	P507
L173	E509
R177	
*R177*	
K171	D511
*K171*	
T63	
S62	
K171	D513
*K171*	

**Table 4 ijms-24-12721-t004:** List of variable loop regions across LC/En, LC/A, LC/F, and LC/X. Crystal structures of LC/En (present structure), LC/A (PDB code:1XTG), LC/F (PDB code:3FIE), and LC/X (PDB code:6F47) were superimposed to identify the equivalent loop regions across the serotypes. The number of residues within the loops is shown, with the loop residue range in brackets.

Loop	LC/En	LC/A	LC/F	LC/X
1	23 (20–42)	13 (21–33)	14 (21–34)	18 (21–38)
2	11 (123–133)	13 (117–129)	15 (117–131)	11 (121–131)
3	6 (172–177)	10 (167–176)	15 (169–183)	7 (169–175)
4	23 (244–266)	18 (242–259)	17 (247–263)	18 (246–263)
5	9 (306–314)	15 (300–314)	14 (303–316)	8 (304–311)

## Data Availability

The atomic coordinates and structure factors have been deposited in the Protein Data Bank under the accession code 8OW8.

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
