# Peer review of "Crystal Structure of the Catalytic Domain of a Botulinum Neurotoxin Homologue from Enterococcus faecium: Potential Insights into Substrate Recognition"

_ijms, 2023, doi:10.3390/ijms241612721_

Round 1
Reviewer 1 Report (New Reviewer)
MANUSCRIPT ID: ijms-2502237
TITLE: Crystal Structure of the Catalytic Domain of a Botulinum Neurotoxin Homologue from Enterococcus faecium: Potential Insights into Substrate Recognition
Dear Editor,
BoNT/En is a homologue of BoNT which is recently identified from Enterococcus faecium. It is the leading cause of hospital acquired multi-drug resistant infections. Understanding how BoNT/En recognizes different substrates is crucial for the future development of inhibitors, antibodies, and potential therapeutics targeting BoNT/En. In this study, the authors presented the crystal structure of light chain BoNT/En at a resolution of 1.8 Å. Through a thorough structural analysis and comparison with previously reported structures, they proposed putative LC/EN subsites, substrate binding sites, and substrate exosite. Also, identified an in vitro autoproteolysis site in the loop 4 region of LC/En. With these structural analysis, they provided valuable insights into the mechanism by which LC/En recognizes its substrates which could aid the development of LC/En inhibitors.
Following are suggestions to strengthen the impact of the manuscript:
Major:
1. In Figure 5 legend, the color coding is confusing. Where is “BoNT/A heavy-chain belt in purple”? Where is LC/X?
Minor:
1. Line 75-76, the message in this sentence “The lack of BoNT/En producing ……” wasn’t very clear. Here is what was cited: “The rarity of BoNT/En producing E. faecium in strains sequenced so far may reflect its recent acquisition, …”.
2. Figure 2: For better presentation, could the author show the “cleft volume” mentioned in Fig2B legend in Fig2A?
3. Table 2: can the author show these putative subsites in a crystal structure figure?
Similarly for Table 3, please use a figure to show potential interactions
Author Response
Reviewer 1
We thank the Reviewer for the constructive comments.
Major:
1. In Figure 5 the color coding is confusing. Where is “BoNT/A heavy-chain belt in purple”? Where is LC/X?
We believe the reviewer is referring to Figure 3 in the original version (Figure 4 in the revised version).
In the original version the heavy chain belt in purple was removed previously from the Figure and appreciate the confusion caused to the Reviewer as the wordings ‘BoNT/A heavy-chain belt in purple’ was left in the Figure legend. In the new figure legend we have removed these wordings have been removed and there should be any confusion.
Regarding LC/X, the colours have been added to the Figure 4 in the revised version..
Minor:
1.Line 75-76, the message in this sentence “The lack of BoNT/En producing ……” wasn’t very clear. Here is what was cited: “The rarity of BoNT/En producing E. faecium in strains sequenced so far may reflect its recent acquisition, …”.
Have added ‘strains’, hopefully improve clarity.
2.Figure 2: For better presentation, could the author show the “cleft volume” mentioned in Fig2B legend in Fig2A?
We are not sure whether showing cleft volume in Figure 2A would enhance any clarity. In fact Figure 2B provides a better quantitative comparison of cleft volumes among different LC molecules studied in this report.
3.Table 2: can the author show these putative subsites in a crystal structure figure?
Similarly for Table 3, please use a figure to show potential interactions.
We have added a figure to show this. Please note that while generating this figure it was noticed that there was an error in subsite identification in the following paper: https://journals.plos.org/plospathogens/article/file?id=10.1371/journal.ppat.1000165&type=printable N388 is actually D388. On inspection of the submitted coordinates it appears the authors meant N368, which is now amended in the table.
Reviewer 2 Report (New Reviewer)
The manuscript entitled "Crystal Structure of the Catalytic Domain of a Botulinum Neu-2 rotoxin Homologue from Enterococcus faecium: Potential In-3 sights into Substrate Recognition " provides an overview about LC/En crystal structure and a detailed analysis in comparison to the previously reported LC/A and LC/F structures in complex with SNAP-25a and a VAMP-like inhibitor. Authors determined the crystal structure of the catalytic domain of a BoNT homologue from Enterococcus faecium (LC/En) at 1.8 Å resolution and provide the substrate recognition mechanism of LC/En. The authors claims that loop 1 is expected to recognize the P end of its substrates and Loop 4 involved in the stabilization of the P end of the substrates. Though I failed to note any mutational evidence which should further corroborate the fact that indeed he loops 1 and 4 offer an insight into substrate binding. Instead, authors tried to validate their data with the available information from the literature. Overall, the abstract is clear and can stand alone. The introduction summarizes the topic well. The detailed, and the analysis looks reasonable for the structure they fitted by molecular replacement. Below are some of my major and minor concerns regarding the article and experiments:
Major Concern:
1. Validation report from RCSB-PDB needs to be attached with the MS (for review purpose) especially when the PDB is on hold. This will ensure better transparency and understanding of the model.
2. It looks, authors stretched data too much to get better resolution, authors tried to push the data to 1.8 Å resolution, having the statistically unacceptable I/sigma(I) = 0.2 and CC1/2 = 14 % values at higher resolution shell. Can authors provide the reason for these statistics? Also it would be great if the authors can provide PDB validation report so that one can look into the validation statistics.
3. Curious whether the authors ever tried to use known homologous structure PDBs as a search model in MR? if yes, what are the statistics for that?
4. Did authors tried to fit any other metal ion at the Zn2+ site and do the refinement on the whole model. If yes what are the statistics for that and how does the omit map looks like.
Minor Concern:
1. Table 1: (Number of atoms), replace protein with amino acids, replace the ions with Zn2+. In place of ligands, can authors provide the name of ligands and their respective no’s.
2. Authors mentioning BoNT structure conserved among its homologues, I failed to note any structure alignment figure and their respective RMSD values?
3. Authors can provide the B-factor of each loop, curious, can authors have detected positive density for the loops? If yes provide the FoFc maps for each loop?
4. Curious, authors ever tried to use HC belt in their crystallization? Or ever tried MD simulation using HC belt, to possibly locate the differences in substrate recognition?
5.

Author Response
Reviewer 2
We thank the Reviewer for the helpful comments.
Major Concern:
1.Validation report from RCSB-PDB needs to be attached with the MS (for review purpose) especially when the PDB is on hold. This will ensure better transparency and understanding of the model.
Have attached the new validation report with the structure now refined at 2Å resolution.
2.It looks, authors stretched data too much to get better resolution, authors tried to push the data to 1.8 Å resolution, having the statistically unacceptable I/sigma(I) = 0.2 and CC1/2 = 14 % values at higher resolution shell. Can authors provide the reason for these statistics? Also it would be great if the authors can provide PDB validation report so that one can look into the validation statistics.
We have now truncated the data to 2Å resolution to match the ‘expected’ values.
3.Curious whether the authors ever tried to use known homologous structure PDBs as a search model in MR? if yes, what are the statistics for that?
MR was successful with LC/X and the AlphaFold generated model. Hence, we did not attempt MR with other models.
4.Did authors tried to fit any other metal ion at the Zn2+site and do the refinement on the whole model. If yes what are the statistics for that and how does the omit map looks like.
We did not model other ions as we performed a zinc edge scan at the synchrotron source confirming the presence of zinc. In addition, the presence of conserved zinc metal binding site residues, it was sufficient to model the catalytic zinc ion.
Minor Concern:
1.Table 1: (Number of atoms), replace protein with amino acids, replace the ions with Zn2+. In place of ligands, can authors provide the name of ligands and their respective no’s.
Have updated Table 1 as suggested by the Reviewer. We did not amend for ligands though, because the crystallisation condition had a mixture of PEG, phosphate etc and we do not think it is necessary to list these.
2.Authors mentioning BoNT structure conserved among its homologues, I failed to note any structure alignment figure and their respective RMSD values?
The proposed analysis by the Reviewer is already known in detail in previously reported publications/s. To avoid repetition, we have cited a reference as opposed to a figure which would seem redundant.
3.Authors can provide the B-factor of each loop, curious, can authors have detected positive density for the loops? If yes provide the FoFc maps for each loop?
Taking into consideration the suggestion made by the Reviewer, we have added B-factors for the loops in Table 1. We have also detailed within the text that regions of loop 2 have poor fit and loop 4 could not be fully modelled. We think that RSRZ outlier values from the validation report and observation of the loops from the deposited structure provide sufficient explanation. We believe extensive electron density figures for these loops are not required.
4.Curious, authors ever tried to use HC belt in their crystallization? Or ever tried MD simulation using HC belt, to possibly locate the differences in substrate recognition?
We did not try to use HC belt in our crystallisation experiments but are planning further experiments in the future. MD simulation was not performed.
Reviewer 3 Report (New Reviewer)
Gregory et al. determined the crystal structure of the catalytic domain of a Botulinum Neu-2 toxin homologue from Enterococcus faecium and compared it with homologous proteins and alphafold models. Despite the authors' claim in the title that this study provides insight into substrate recognition, most of the structural analyses primarily focus on highlighting structural differences from other homologous proteins. Biochemical analysis of the autoproteolysis of this protein was performed and compared to similar proteins. While I anticipate that this study could contribute to the advancement of this field and related areas, I must also point out that it lacks novelty and insightful components. Overall, I believe there is ample room for scientific improvement in this manuscript.
1. Structure comparison can provides valuable insights into evolution. The variations in loop structures can be understood through the sequence of amino acids, and it may seem questionable for the authors to describe them in extensive detail. However, a more comprehensive interpretation of the structure of the catalytic domain of a Botulinum Neu-2 toxin homologue from Enterococcus faecium, as determined by the authors, is needed. I have encountered difficulty in identifying any substantial insights regarding the claimed substrate recognition in the title.
2. While the Alphafold2 model serves as a valuable reference tool for gaining insights into overall protein function, the manuscript appears to have an excessive reliance on references to the Alphafold2 model. Additional experiments are necessary to validate and support the findings based on the Alphafold2 predicted model. It is important to note that although Alphafold2 is a groundbreaking technique, it may provide information, such as conformational details, that do not always match with experimental results. Therefore, the authors should place greater emphasis on interpreting the structures obtained through experimental result. This shift in focus would not only enhance the development of the manuscript but also contribute significantly to related fields.
3. Figure 2B, Table2, Table 3 and Table 4 are not at all intuitive and difficult to understand.
4. I think that the highest shell values of <I/sigma(I)> and CC1/2 in Table 1 are lower than the general standards. I think author should probably lower the resolution to meet the general standards. What was the resolution cutfoff criterion in particular?
5. The figure legend should be more concise, and it is suggested to remove unnecessary text.
- Authors should consider whether displaying conformation using an alphafold model is scientific.
-'The blue box indicates regions of conserved similarity. Sequence align-250 ments were produced using Clustal Omega [52] and ESPript [53].' should move to method section
6. There should be no spaces between amino acids and amino acid numbers.
Readability was not high, and sentence redundancy was high.
Author Response
Reviewer 3
Authors- We thank the Reviewer for critical reading of our manuscript and for the comments.
- Structure comparison can provide valuable insights into evolution. The variations in loop structures can be understood through the sequence of amino acids, and it may seem questionable for the authors to describe them in extensive detail. However, a more comprehensive interpretation of the structure of the catalytic domain of a Botulinum Neu-2 toxin homologue from Enterococcus faecium, as determined by the authors, is needed. I have encountered difficulty in identifying any substantial insights regarding the claimed substrate recognition in the title.
Reply- We appreciate the point made by the Reviewer. Based on the existing data and bioinformatic tools available to us, we have made a sincere attempt to gain some insight into substrate recognition from the structure as described in our present manuscript. Science is a progressive field and our hypothesis need further validation with additional biological experiments which is beyond the scope of our present manuscript.
- While the Alphafold2 model serves as a valuable reference tool for gaining insights into overall protein function, the manuscript appears to have an excessive reliance on references to the Alphafold2 model. Additional experiments are necessary to validate and support the findings based on the Alphafold2 predicted model. It is important to note that although Alphafold2 is a groundbreaking technique, it may provide information, such as conformational details, that do not always match with experimental results. Therefore, the authors should place greater emphasis on interpreting the structures obtained through experimental result. This shift in focus would not only enhance the development of the manuscript but also contribute significantly to related fields.
Reply- The AlphaFold model was used along with the experimentally determined structure to offer insight into substrate binding through modelling of the belt into the experimentally determined structure based on structural alignments. Additionally, BoNT/A and BoNT/F structures in complex with their substrate/inhibitors were also used for comparison with the LC/En structure to aid in the identification of the putative subsites (as listed in Table 2) and exosites (as listed in Table 3). Additional emphasis on the experimentally determined crystal structure in relation to substrate recognition requires further experiments which is beyond the scope of our present manuscript. Our structural comparison identifies only putative subsites/exosites. To address this point in line with the Reviewer comment, we have added a sentence in the conclusion section of the manuscript.
- Figure 2B, Table2, Table 3 and Table 4 are not at all intuitive and difficult to understand.
Reply- We are unsure about this dismissive comment. The other two Reviewers did not seem to have a problem with these figures. We have modified Table 2.
- I think that the highest shell values of <I/sigma(I)> and CC1/2 in Table 1 are lower than the general standards. I think author should probably lower the resolution to meet the general standards. What was the resolution cutoff criterion in particular?
Reply- Data has been truncated to 2Å resolution to improve the statistics as suggested by the Reviewer.
- The figure legend should be more concise, and it is suggested to remove unnecessary text.
- Authors should consider whether displaying conformation using an alphafold model is scientific.
-'The blue box indicates regions of conserved similarity. Sequence alignments were produced using Clustal Omega [52] and ESPript [53].' should move to method section.
Reply- It is debated currently in the field that a difference between the AlphaFold model and crystallographic model may be due to crystal contacts capturing ‘one’ of the possible dynamic states and that the differences in the AlphaFold model may be offering insight into an ‘alternative’ conformation for a dynamic protein sample. This validates the rationale behind comparison of our experimentally determined crystal structure to that of the predicted. We also clearly state that this difference ‘provides some insight into potential dynamics’ in the manuscript which requires further experimental validation.
It is unclear which figure the Reviewer is referring to but it is assumed that wither Figure 3 or 5 given the mention of sequence alignment. We have removed the ‘sequence alignment...’ section and added it to the methods to reduce unnecessary wordings and condensed it which is already described within the main text.
- There should be no spaces between amino acids and amino acid numbers.
Reply- Have amended as suggested by the Reviewer.
Round 2
Reviewer 2 Report (New Reviewer)
I have went through the manuscript and noted all the queries that I have asked were responded and edited into the manuscript. Therefore I recommend this article for publication based on the response received from queries.
This manuscript is a resubmission of an earlier submission. The following is a list of the peer review reports and author responses from that submission.
Round 1
Reviewer 1 Report
The present manuscript describes the tertiary structure of the catalytic domain of a botulinum neurotoxin like molecule from Enterococcus faecium (BoNT/En). The established structure (LC/En) was superimposed with a predicted structure for the full-length BoNT/E and previously published catalytic domain structures of botulinum neurotoxin serotypes A (LC/A) bound to its substrate SNAP-25 and serotype F (LC/F) bound to a substrate derived inhibitor as well as full-length BoNT/A. These comparisons were done to analyze the putative binding mode of the three SNARE substrates and predict the enzyme subsites in LC/En. In addition, a putative autoproteolytic cleavage site is reported for LC/En.
The manuscript is well organized and the clearly written. However, figure 3A, C, D and figure 4A and C are not really instrumental in understanding text statements. The panels are somewhat overloaded (Fig. 3A, C, D), but are lacking details about enzyme subsites discussed in the text (3A, C, D and 4A, C). It might be better to show individual comparisons and close-up views of individual predicted enzyme subsites (partly as supplementary figures).
Together, the study provides the structure of another member of the zinc dependent protease family of clostridial neurotoxins and neurotoxin-like molecules of other species. A weakness of the manuscript is that the main part of the manuscript discusses the putative substrate binding mode of the enzyme, but experimental validation is missing for any of those assumptions. Thus, the reported structure represents a basis for subsequent mutational analyses to understand how this enzyme manages cleavage of members of all three SNARE protein families.
minor comments:
line 62, ref. 33 is inappropriate for SNAP-25 cleavage by BoNT/C. Blasi et al., 1993 (EMBO J 12, 4821-4828) first showed cleavage of syntaxin and Foran et al., 1996 (Biochemistry 35, 2630-2636), Osen-Sand et al., 1996 (J Comp Neurol 367, 222-234) or Williamson et al., 1996 (J Biol Chem 271, 7694-7699) cleavage of SNAP-25.
line 69, ref. 35 should also be checked. Ref. 35 reports the genome sequence of Weissella Oryzae SG25T, however the ORF encoding a botulinum neurotoxin like protein was discovered by Mansfield et al., 2015 (FEBS Lett 589, 342–348).
line 83, delete irrelevant text insertion
lines 86, 87, unlike the cleavage of SNAP-25 by BoNT/A, C, and E, cleavage of SNAP-25 and SNAP-23 by BoNT/En does not occur at the C-terminal end!
lines 149-151, is it meaningful to compare the putative LC/En subsites with LC/A subsites? The substrate residues P1 to P5’ binding to subsites S1 to S5’ are quite dissimilar, QRATKM (LC/A) versus KDMKEA (LC/En).
line 172, the text specifies 6 potential salt bridges between LC/En and the heavy chain derived ‘belt’ region based on superposition of the solved LC/En structure and the modelled full-length structure. However, Tab. 3 specifies only 3 such interactions, LC-R177 to belt-E509, LC-K171 to belt-D511, and LC-K171 to belt-D513 (shown in italics). A fourth one might be LC-R119 to belt-D488.
lines 225 and 237, “Table 2” should read “Table 3”
Author Response
The present manuscript describes the tertiary structure of the catalytic domain of a botulinum neurotoxin like molecule from Enterococcus faecium (BoNT/En). The established structure (LC/En) was superimposed with a predicted structure for the full-length BoNT/E and previously published catalytic domain structures of botulinum neurotoxin serotypes A (LC/A) bound to its substrate SNAP-25 and serotype F (LC/F) bound to a substrate derived inhibitor as well as full-length BoNT/A. These comparisons were done to analyze the putative binding mode of the three SNARE substrates and predict the enzyme subsites in LC/En. In addition, a putative autoproteolytic cleavage site is reported for LC/En.
Authors’ reply- We thank the Reviewer for the constructive comments about our manuscript.
The manuscript is well organized and the clearly written. However, figure 3A, C, D and figure 4A and C are not really instrumental in understanding text statements. The panels are somewhat overloaded (Fig. 3A, C, D), but are lacking details about enzyme subsites discussed in the text (3A, C, D and 4A, C). It might be better to show individual comparisons and close-up views of individual predicted enzyme subsites (partly as supplementary figures).
Authors’ reply-We think Figure 3A is necessary to illustrate the proximity of each loop relative to one another and the difference in conformation of each loop relative to BoNT/A, BoNT/F and BoNT/En (full-length alphafold prediction) as in the text we state ‘we have identified a total of 5 loops (designated as loops 1-5) that have been associated with substrate and belt recognition in the LC/En, LC/A:SNAP-25, and LC/F:VAMP2 structures. Relative to the scissile bond, LC/A loops 1, 2, 3 and 5 bind the N-terminal portion of SNAP-25, whereas loop 4 recognises the C-terminal portion [45]. These loops vary in both length (Table 4) and conformation (Figure 3A) across the serotypes accounting for variation in substrate binding.’
Figure 3C has been amended for clarity, removing the loops of /A and /F as they are not necessary and the difference in conformation is already illustrated in Figure 3A. This highlights the distance of loop 1 relative to SNAP-25, VAMP2, and the En belt (alphafold model) as stated in the text.
Figure 3D has also been amended for clarity. We have reoriented the view and removed /F loops to make it clear. The observations made for BoNT/A to BoNT/En are similar to that of BoNT/F so we have amended the figure legend and state that we observe similar features for BoNT/F but that it has been omitted from the figure for clarity.
We agree with the Reviewer that there is some repetition of information presented here. We have changed Figure 4 by removing 4A and replacing it with 4C. We think 4C is necessary as we state that loop 4 is close to the active site zinc ion and this figure illustrates the proximity of the terminus of the substrate to loop 4, which supports our statement ‘It is, therefore, probable that loop 4 serves a similar function in LC/En, stabilising the P’ end of SNAP-25 and VAMP, due to its location relative to the active site zinc ion’. We also mention how in LC/A loop 4 stabilises the substrate in the text by beta-stacking interactions which this figure shows.
Together, the study provides the structure of another member of the zinc dependent protease family of clostridial neurotoxins and neurotoxin-like molecules of other species. A weakness of the manuscript is that the main part of the manuscript discusses the putative substrate binding mode of the enzyme, but experimental validation is missing for any of those assumptions. Thus, the reported structure represents a basis for subsequent mutational analyses to understand how this enzyme manages cleavage of members of all three SNARE protein families.
Authors’ reply- We are thankful to the Reviewer for a positive comment and believe that the language we have used reflects that these are insights based purely on the available structural data and literature (for other serotypes). Direct experimental data for BoNT/En is required (which is beyond the scope of this manuscript) but our manuscript serves as a basis for future experimental design.
Minor comments:
line 62, ref. 33 is inappropriate for SNAP-25 cleavage by BoNT/C. Blasi et al., 1993 (EMBO J 12, 4821-4828) first showed cleavage of syntaxin and Foran et al., 1996 (Biochemistry 35, 2630-2636), Osen-Sand et al., 1996 (J Comp Neurol 367, 222-234) or Williamson et al., 1996 (J Biol Chem 271, 7694-7699) cleavage of SNAP-25.
Authors’ reply- We thank the Reviewer for pointing out these omissions. We have added two of these references at appropriate places in the text.
line 69, ref. 35 should also be checked. Ref. 35 reports the genome sequence of Weissella Oryzae SG25T, however the ORF encoding a botulinum neurotoxin like protein was discovered by Mansfield et al., 2015 (FEBS Lett 589, 342–348).
Authors’ reply- Our statement is on how the gene was isolated, hence we think the reference chosen is appropriate here, however we have added the additional reference as well.
line 83, delete irrelevant text insertion
Authors’ reply- Deleted
lines 86, 87, unlike the cleavage of SNAP-25 by BoNT/A, C, and E, cleavage of SNAP-25 and SNAP-23 by BoNT/En does not occur at the C-terminal end!
Authors’ reply- We thank the Reviewer for pointing out this error! We have corrected this to state that it is the N-terminal portion.
lines 149-151, is it meaningful to compare the putative LC/En subsites with LC/A subsites? The substrate residues P1 to P5’ binding to subsites S1 to S5’ are quite dissimilar, QRATKM (LC/A) versus KDMKEA (LC/En).
Authors’ reply- We think it is interesting to note that although they bind different substrates/the same substrate at different locations, they share similar properties within the subsites. Have reworded this sentence to clarify that point.
line 172, the text specifies 6 potential salt bridges between LC/En and the heavy chain derived ‘belt’ region based on superposition of the solved LC/En structure and the modelled full-length structure. However, Tab. 3 specifies only 3 such interactions, LC-R177 to belt-E509, LC-K171 to belt-D511, and LC-K171 to belt-D513 (shown in italics). A fourth one might be LC-R119 to belt-D488.
Authors’ reply- PDBePISA lists salt bridging interactions by atom (e.g R177 is listed 3 times as a salt bridge depending on which atom of Glu 509 it interacts with). We have amended the number of salt bridging interactions in text. K171 appears twice in the table in italics and the figure legend has been amended to explain this.
lines 225 and 237, “Table 2” should read “Table 3”
Authors’ reply- Yes indeed. Thank you. We have corrected this error.
Reviewer 2 Report
The paper describes the Crystal Structure of the Catalytic Domain of a Botulinum Neurotoxin Homologue from Enterococcus faecium. Although it is compared with other types in terms of structure, site and sequence, its Potential sights into Substrate Recognition are not well explored. More content and ideas are conjectures that require more experimental data to support the paper's ideas, or experimental results to support the results based on the crystal structure. In addition, the writing strategy of the article may result in this condition that the not directly related experts can not read the content and details of the article. Overall, the paper is some value, but the content need redefined and suppled, the depth of the article needs to be improved. There TWO major questions still need to be addressed. One is the question of figure 6. In figure 6, the immunoblotting did not show a signal for the ~50 kDa intact LC/En (containing an N-terminal His tag) and the ~30 kDa N-terminal fragment, but not the ~20 kDa C-terminal fragment of LC/En. In addition, it is not scientific, or only speculative, to determine by size whether the self-cutting site is in Loop 4. The authors can determine the self-cleavage sites of these two fragments by determining their N and C residue sequences.Another is the results of Table 2. The LC/En cleaves VAMP1, VAMP2 and VAMP3 beween an Ala and Asp residue (Ala 69 and Asp 70 for VAMP1, Ala 67 and Asp 68 for VAMP2, and Ala 54 and Asp 55 for VAMP3). The LC/En cleaves SNAP-25 beween K63 and D64. A conserved HExxH zinc coordinating motif Click or tap is the Active Site of BoNT/en. However, 2.2 The Active Site of this paper describe the Putative LC/En subsites, these content is related with above HExxH or not. In short, this part of the content, I do not understand, or the way the author described is not correct.

Author Response
The paper describes the Crystal Structure of the Catalytic Domain of a Botulinum Neurotoxin Homologue from Enterococcus faecium. Although it is compared with other types in terms of structure, site and sequence, its Potential sights into Substrate Recognition are not well explored. More content and ideas are conjectures that require more experimental data to support the paper's ideas, or experimental results to support the results based on the crystal structure. In addition, the writing strategy of the article may result in this condition that the not directly related experts can not read the content and details of the article. Overall, the paper is some value, but the content need redefined and suppled, the depth of the article needs to be improved.
Authors’ reply- This is a structural biology based manuscript. We are unsure what the Reviewer meant here in his/her statement. In particular the language used by the Reviewer is difficult makes it difficult to understand the point/s. We also note that the Reviewer has stated that – ‘ I am not qualified to assess the quality of English in this paper’, which is an unusual statement !
There TWO major questions still need to be addressed. One is the question of figure 6. In figure 6, the immunoblotting did not show a signal for the ~50 kDa intact LC/En (containing an N-terminal His tag) and the ~30 kDa N-terminal fragment, but not the ~20 kDa C-terminal fragment of LC/En. In addition, it is not scientific, or only speculative, to determine by size whether the self-cutting site is in Loop 4. The authors can determine the self-cleavage sites of these two fragments by determining their N and C residue sequences.
Authors’ reply- We did not determine the location of the cleavage site based on the sizes, there are conserved cleavage sites previously identified for BoNT/A, /B, and /E within the equivalent loop 4 of these serotypes. If it did cut here, it would result with fragment sizes of 20 and 30kDa, which is what we have observed on the gel picture provided. We state that ‘these results support the premise that LC/En undergoes autoproteolysis’. The Reviewer states that the immunoblotting did not show the 50 kDa intact LC/En and the 30 kDa N-terminal fragment. This is contradictory to what we state in the manuscript. We clearly state that ‘immunoblotting does show the 50 and 30 kDa fragment, but not the 20 kDa fragment’. This is not used as direct evidence for the autoproteolytic site within loop 4, it is used in combination with literature for BoNT/A, /B and /E and evidence of statements as stated above.
Another is the results of Table 2. The LC/En cleaves VAMP1, VAMP2 and VAMP3 beween an Ala and Asp residue (Ala 69 and Asp 70 for VAMP1, Ala 67 and Asp 68 for VAMP2, and Ala 54 and Asp 55 for VAMP3). The LC/En cleaves SNAP-25 beween K63 and D64. A conserved HExxH zinc coordinating motif Click or tap is the Active Site of BoNT/en. However, 2.2 The Active Site of this paper describe the Putative LC/En subsites, these content is related with above HExxH or not. In short, this part of the content, I do not understand, or the way the author described is not correct.
Authors’ reply- We have added additional information to the subheading to make it clear that this section is about the active site and the extended cleft which makes up the subsites that contribute to substrate recognition and cleavage.
Round 2
Reviewer 1 Report
All points of criticism were adequately addressed.
Clarity of Figures 3C and 3D increased significantly by the amendments.
Author Response
We are pleased to note that this Reviewer is satisfied with the revisions we have made in the revised version.
Thank you.
Reviewer 2 Report
Without modification marks and detailed modification instructions, the reviewer cannot find differences and improvements between the current revision and the previous submission. I'm not sure how the current revision improves on the previous submission. Moreover, the revised paper did not answer my question. More importantly, the main weakness of the manuscripts is that their results and claims are mostly speculative or reference to other people's data, without their own experimental data, and lack of experimental validation to support these hypotheses and results. In their paper, they clearly state that ‘immunoblotting does show the 50 and 30 kDa fragment, but not the 20 kDa fragment’. However, in figure 6 or somewhere else, there are not any the immunoblotting results. The result of the experiment is the most basic. If the results of this experiment cannot be provided or do not want to be provided, how to make the review experts believe your data and results. In addition, it is not scientific, or only speculative, to determine by size whether the self-cutting site is in Loop 4. The authors can determine the self-cleavage sites of these two fragments by determining their N and C residue sequences. Simply quoting someone else's data is not persuasive. In fact, this experiment is also very simple.Without modification marks and detailed modification instructions, the reviewer cannot find differences and improvements between the current revision and the previous submission. I'm not sure how the current revision improves on the previous submission. Moreover, the revised paper did not answer my question. More importantly, the main weakness of the manuscripts is that their results and claims are mostly speculative or reference to other people's data, without their own experimental data, and lack of experimental validation to support these hypotheses and results. In their paper, they clearly state that ‘immunoblotting does show the 50 and 30 kDa fragment, but not the 20 kDa fragment’. However, in figure 6 or somewhere else, there are not any the immunoblotting results. The result of the experiment is the most basic. If the results of this experiment cannot be provided or do not want to be provided, how to make the review experts believe your data and results. In addition, it is not scientific, or only speculative, to determine by size whether the self-cutting site is in Loop 4. The authors can determine the self-cleavage sites of these two fragments by determining their N and C residue sequences. Simply quoting someone else's data is not persuasive. In fact, this experiment is also very simple.
Author Response
We appreciate the point made the Reviewer.
In order to provide clarity, we have added the western blot (revised Figure 6A with the modified legend and also in the main text of the manuscript lines 331, 334, 376-379). The western blot was produced using anti-poly-histidine antibody revealing only Lc/En fragments with a His-tag. The western blot shows bands for 50 kDa, 34 kDa, 30 kDa, and 15 kDa. The SDS-PAGE shows clear bands at 50 kDa, 30 kDa and 20 kDa. The 20 kDa band is absent from the western blot and is therefore the C-terminus, with the N-terminus being the 30 kDa band. The western blot also indicates further protease/degradation sites present. However, the dominant cleavage site would be that which results in the 30 kDa and 20 kDa as evidenced by the SDS-PAGE.
We very much hope this will address the main concern from this Reviewer.
Round 3
Reviewer 2 Report
The revised paper still did not completely answer my question. In revised figure 6, the western blot shows bands for 50 kDa, 34 kDa, 30 kDa, and 15 kDa. The 50 and 30 kDa fragment are correct, but what are the 34 kDa and 15 kDa fragment on earth. The 34 kDa and 15 kDa fragment contains the his tag, which indicates there is other cutting or degradation site. In addition, it is not scientific, or only speculative, to determine by size whether the self-cutting site is in Loop 4. The authors can determine the self-cleavage sites of these two fragments by determining their N and C residue sequences. Simply quoting someone else's data is not persuasive. In fact, this experiment is also very simple and should been finished by author.
